# Study on the Effects of Grouting and Roughness on the Shear Behavior of Cohesive Soil–Concrete Interfaces

**DOI:** 10.3390/ma13143043

**Published:** 2020-07-08

**Authors:** You-Bao Wang, Chunfeng Zhao, Yue Wu

**Affiliations:** 1Department of Geotechnical Engineering, Tongji University, Shanghai 200092, China; tjzhchf@tongji.edu.cn; 2Laboratory of Geotechnical and Underground Engineering of Ministry of Education, Tongji University, Shanghai 200092, China

**Keywords:** grouting volume, cohesive soil–concrete interface, roughness, direct shear test, shear band

## Abstract

Grouted soil–concrete interfaces exist in bored piles with post-grouting in pile tip or sides and they have a substantial influence on pile skin friction. To study the effect of grouting volume on the shearing characteristics of the interface between cohesive soil and concrete piles with different roughness, grouting equipment and a direct shear apparatus were combined to carry out a total of 48 groups of direct shear tests on cohesive soil–concrete interfaces incorporating the grouting process. The test results showed that the shear behavior of the grouted cohesive soil–concrete interface was improved mainly because increasing the grouting volume and roughness increased the interfacial apparent cohesion. In contrast, increasing the grouting volume and roughness had no obvious increasing effects on the interfacial friction angle. Interfacial grouting contributed to the transition in the grouted cohesive soil from shrinkage to dilation: as the grouting volume increased, the shrinkage became weaker and the dilation became more obvious. The shear band exhibited a parabolic distribution rather than a uniform distribution along the shearing direction and that the shear band thickness was greater in the shearing direction, and it will become thicker with increasing grouting volume or roughness. The analysis can help to understand the shear characteristics of soil–pile interface in studying the vertical bearing properties of pile with post-grouting in tip or sides.

## 1. Introduction

Grouting technology, which provides good mechanical property improvements, is widely used in engineering applications, such as tunnel engineering, pile foundation engineering, and construction [1]. In the ultralong, bored, cast-in-place piles used in high-rise buildings, the slurry used to prevent hole wall collapse and the mud sedimentation at the bottom of the pile decrease pile shaft friction, which substantially decreases the bearing capacity of the pile [2]. To improve the bearing capacity of a pile, post-grouting on the tips and sides of piles can increase the tip friction and the shaft friction of piles, respectively. Many field tests and numerical analyses have studied the vertical bearing performance of grouted piles [2,3,4,5,6,7]; however, research on the shear characteristics of the grouted soil–pile interface in grouted piles is relatively rare.

Following the work of Potyondy [8], who studied the interface between soil and various construction materials, many scholars have studied the characteristics of the soil–structure interface [9,10,11,12,13,14,15,16,17,18,19,20,21,22,23,24,25,26,27,28,29,30,31,32,33]. These studies included apparatus modifications and new test methods [9,10,11], which were used to investigate the influence of surface properties on the traditional soil–structure interface (including soil–concrete, soil–steel, and soil–cement grout interfaces) [9,12,13,14,15,16,17,18,19,20,21,22,23,24,25,33,34], the interfaces between new materials (including ice–soil and soil–geogrid interfaces) [26,27,28,29,30,31,32], and the exterior load conditions (stress history, monotonic load, or cyclic load condition) [35,36,37].

In particular, for the soil–pile interface, there are some developments that need further elaboration in terms of the influence of the surface property on the traditional soil–structure interface.

Some studies have investigated the influence of roughness or soil type on the shear characteristics of the soil–concrete interface. Hu and Pu [9] conducted a study of shear tests of Yongdinghe sand–steel interface with different roughnesses, finding that elastic perfect-plastic failure mode occurs along the smooth interface while strain localization occurs in a rough interface accompanied with strong strain-softening and bulk dilatancy; Su and Zhou [13] carried directed shear tests of sand–steel interface and found that a critical value of relative roughness exists such that the peak shear stress or friction angle can no longer be readily enlarged when relative roughness exceeds it. Zhao [17] found that the thickness of shear zone is 8–14 times average particle size in the sand–concrete interface shearing test. Chen [21] conducted a series of laboratory large-scale direct shear tests using different types of red clay–concrete interfaces and the surface roughness is found to have a remarkable effect on the interfacial shear strength and shear behavior, which is that the shear strength increases with increased surface roughness. Shakir and Zhu [24] carried out direct shear tests to study the effects of water content and surface roughness on the shear stress–shear displacement relationship of a clay–concrete interface, wherein they found that the interfacial shear sliding dominated the interfacial shear displacement behavior for both relatively rough and smooth interfaces.

Many other experimental studies have investigated the characteristics of the interface between soil and cement grout. Chen [20] performed many direct shear tests and found that there is a linear relationship between the shear strength of the grout and resultant normal pressure. Hossain [18] conducted a series of interface direct shear tests between compacted, completely decomposed granite (CDG) soil and cement grout under saturated conditions with different grouting pressures and normal stresses, finding that grouting pressure and normal stress have influence on the behavior of the soil–cement interface. Chu and Yin [33] conducted direct shear tests between the CDG and a cement grout plate, finding that the shear stress–displacement behavior of the soil–cement grout interface is similar to that of the soil alone and the interface shear strength of the CDG and cement grout material depends on the normal stress level, the soil moisture content, and the interface surface waviness.

Previous research has focused on the soil–concrete interface or soil–cement grout interface, and the grouting process used in the test specimen preparation process has not been taken into account. In fact, regardless of the tunnel engineering or post-grouting processes of piles, the cement grout cannot completely cover the interface between the soil and concrete. The previous research performed by our research group also shows that cement grout cannot completely cover the interface because of the different grouting modes under different loading conditions [38]. Grouting will affect the roughness and subsequently the mechanical properties of the soil–pile interface. Therefore, it is necessary to consider the grouting process and roughness in the soil–concrete interface.

There have also been some numerical simulations and digital image analyses of the shear deformation during the interfacial shearing process [17,39,40,41]. Zhao [17] used the discrete element method (DEM) to simulate the interface direct shear behavior of granular soil and rough surface, and discovered the micro-mechanisms of the interface shear band deformation. Zhang [40] also simulated the granular-continuum interfaces using DEM and investigated the micro-scale responses of the interface simulations. Oda [41] used microfocus X-ray computed tomography to observe the microstructure in shear band. However, these methods either need to define complex conditions, or require a lot of calculations, or require expensive and complicated instruments, and intuitive experimental observations and measurements of the shear band at the macroscale are few. Therefore, a simple method of punching bars of colored particles into soil was designed to observe and measure interfacial shear band thickness and distribution.

Thus, in order to study the effect of grouting volume on the shearing characteristics of the interface between cohesive soil and concrete piles with different roughness, the multifunctional direct shear apparatus in the laboratory was modified and combined with the self-developed grouting device to carry out 48 sets of direct shear tests on grouted cohesive soil–concrete interfaces considering the grouting process. The shear performance of the grouted cohesive soil–concrete interface with different roughness values was analyzed from four aspects: the shear stress–displacement relationship, interfacial shear strength, dilatancy, and shear band thickness. The results showed that grouting and roughness increased the shear strength and shear stiffness mainly by increasing the interfacial apparent cohesion rather than the friction angle, and grouting and roughness can weaken the shear shrinkage and increase the dilatancy of the cohesive soil. The increase of shear strength was also reflected in the increase of the shear band thickness with increasing grouting volume and roughness. These results from the grouted cohesive soil–concrete interfaces provide experimental support for understanding and analyzing the vertical bearing characteristics of piles with post-grouting in tip or sides.

## 2. Apparatus, Materials, and Methods

### 2.1. Apparatus for Grouting and Direct Shear Test

The direct shear test was performed on a large-scale multifunctional interfacial shearing instrument at the Key Laboratory of Geotechnical and Underground Engineering of the Ministry of Education, Tongji University (Shanghai, China); some modifications were made to the instrument to ensure that the grout could be injected into the soil–concrete interface. The main equipment was divided into two parts: grouting equipment and an SJW-200 direct shear apparatus, as shown in Figure 1. The parameters of the SJW-200 direct shear apparatus are listed in Table 1.

### 2.2. Preparation of Materials

#### 2.2.1. Concrete Plate

There are many kinds of roughness definitions, but for convenience, the peak-to-valley distance R proposed by Zhang [42] was taken to define the roughness of the interface, as shown in Figure 2. To investigate the effect of roughness on grouted cohesive soil–concrete plate interfaces, three roughness values were selected in this experiment: 0, 3, and 6 mm. A concrete plate with dimensions of 600 × 400 × 50 mm (length × width × height) was prepared, as shown in Figure 3, among which S_2_ = S_3_, the height of equilateral trapezoidal tooth (h) equals R. To overcome the brittleness of the concrete, 6 mm of steel-reinforced mesh was arranged at 100 mm intervals near the lower and upper surfaces.

#### 2.2.2. Cohesive Soil and Grout

In this test, samples of remolded soil were prepared manually. The source of the soil was the silty clay layer ⑧ taken from a project in Shanghai. Due to the large amount of soil used, the original soil was first naturally dried, crushed, dried again, and then passed through a 0.1 mm sieve to obtain dry soil. Cohesive soil samples were prepared from this dry soil in accordance with the [41]; the parameters of the soil samples are shown in Table 2.

The cement used for grouting was 32.5R Portland cement from Hailuo Shanghai Cement Co., Ltd. (Shanghai, China) The parameters of the cement are shown in Table 3. According to a preliminary test, the injectability, fluidity, and setting time of the cement grout were appropriate when the water-to-cement ratio was set to 0.6. The grouting pressure was set to 0.4 MPa. The water used was tap water. An early strength accelerator (0.5% of the cement content) was mixed in the grout to accelerate the solidification process and reduce the curing time. Very little red stain was incorporated to facilitate later observation.

### 2.3. Test Methods and Scheme

#### 2.3.1. Measurement of Shear Band

To observe the changes in the cohesive soil at the interface during the shearing process and effectively measure the maximum depth at which the shearing process influences the soil (i.e., the shear band thickness), bars of fine red particles were punched into the soil samples. The colored bar layout is shown in Figure 4. The schematic diagram in Figure 5 shows the mutual positions of shear box displacement *S* (i.e., shear displacement), relative shear displacement *S_X_* and shear band thickness *S_Z_* during the shearing process under the normal force *F_V_* and shear force *F_S_*. After the shearing process is completed, the soil can be excavated to perform a detailed measurement of the shear band.

#### 2.3.2. Test Scheme

The tests were divided into three categories according to the concrete plates with roughness values of 0, 3, and 6 mm, namely, R0, R3, and R6. There were 16 specimens in each category, for a total of 48 specimens for R0, R3, and R6. Only the 16 specimens in category R0 are listed in Table 4 and the specimens in the other two categories have the same grouting and loading conditions.

#### 2.3.3. Test Procedures

After the concrete plate and soil sample are in place in the shear box, the loading and grouting process should be carried out according to the following procedures, as illustrated in Figure 6 (R0-V2-C100-S75 is taken as an example):
Loading: Apply an initial normal consolidation stress of 100 kPa to the soil until the vertical deformation of the cohesive soil becomes stable. The consolidation time should be no less than 1 h.Grouting and curing: Reduce the normal stress to 75 kPa, ensure the soil continues to consolidate for 1 h. During this time, connect all the equipment, prepare the 0.2 dm^3^ of grout, and inject the grout into the interface at 0.4 MPa. After injecting the grout, keep the normal stress at 75 kPa for consolidation for 10 min. Then push the shear box, which is full of grouted soil, away from the apparatus platform, cover it with a plastic film and store it under a thick wood plate to cure for 5 days.Loading and shearing: After curing the grout, install the shear box in place, apply a normal stress of 100 kPa to the soil for 1 h, and then reduce the normal stress to 75 kPa for 30 min; the shearing should be set to 1 mm/min until the shear displacement reaches 40 mm.Cutting, observation, and measurement: After shearing, remove the shear box from the apparatus, cut the soil along the rows of colored bars, observe the apparent soil characteristics, and measure the shear band.


## 3. Test Results and Discussions

By conducting the above tests, the effects of roughness, grouting volume, and normal loading and unloading on interfacial shear performance could be obtained. Due to the limited length of this article, the specific effects of roughness and grouting volume on the shear characteristics of the interface between cohesive soil and concrete plate were mainly analyzed from the aspects of interfacial shear stress–displacement curves, shear strength, dilatancy, and shear band thickness and distribution. 

### 3.1. Roughness Influence

#### 3.1.1. Shear Stress–Displacement Relationship

Figure 7 shows the relationship between the interfacial shear stress–displacement curves (*τ*–*s* curves) and roughness. Generally, each *τ*–*s* curve presents three stages. In the early stage, there is a steep straight curve, which gradually becomes flatter in the middle stage and stabilizes in the later stage. The whole shearing process shows hardening behavior. Although the increase amplitude is different under various loading conditions, the peak shear stress gradually increases as the roughness increases from R0 to R6. As the roughness increases, the initial straight stage of the *τ*-*s* curve becomes steeper, indicating that increasing roughness can increase the initial shear stiffness. Figure 7e shows a slight abnormality in that the curve of specimen R3-V0-C75-S75 is higher than that of specimen R6-V0-C75-S75 during the early stage, which may be due to more grout being on the interface rather than in the soil during shearing. Figure 7f shows a slight drop in the curve of R6-V2-C50-S50 during the middle stage, which may be ascribed to grout slipping at the interface.

#### 3.1.2. Interfacial Shear Strength

With reference to the Mohr–Coulomb criterion, the peak shear stress and normal stress of the interface conform to the equivalent relationship as
(1)$$\tau =c_{e}+\sigma tan\varphi_{e}$$
where $c_{e}$ and $\varphi_{e}$ represent the equivalent apparent cohesion and the equivalent friction angle, respectively.

The increase of the shear strength is specifically reflected in the changes of $c_{e}$ and $\varphi_{e}$. The normal stress and the peak shear stress are linearly fitted under different roughness values according to Equation (1), as shown in Figure 8. The parameters of the fitted curves are listed in Table 5. The apparent cohesion increases by 1.25 times from 4.61 kPa at R0 to 10.4 kPa at R6 without grouting, whereas the friction angle only increases by about 10% from 31.42° to 34.52°. Under the V2 grouting condition, the apparent cohesion increases from 10.83 kPa at R0 to 19.39 kPa at R6, whereas the friction angle only increases from 32.94° to 36.05°. Figure 8 and Table 5 clearly show that, under the same grouting volume, roughness can significantly increase the apparent cohesion of the grouted cohesive soil–concrete interface, whereas it has no obvious influence on the friction angle.

#### 3.1.3. Shear Dilatancy

Figure 9 shows the influence of roughness on the dilation or shrinkage (in the form of the relationship between vertical displacement and horizontal displacement, wherein a positive vertical displacement indicates soil shrinkage and a negative vertical displacement means dilation) during interfacial shearing. Without grouting, all cohesive soil samples exhibit shear shrinkage. With decreasing roughness, the soil exhibits weakening shrinkage. As shown in Figure 9c, the shearing process is not very stable, wherein the vertical displacement decreases in the middle stage; however, in general, the greater the roughness is, the weaker the shear shrinkage of the soil. Table 6 shows the final shrinkage values and their relative variation as the roughness changes from R0 to R3 and from R3 to R6. The negative variation values indicate that the cohesive soil shrinkage decreases with increasing roughness.

### 3.2. Grouting Volume Influence

#### 3.2.1. Shear Stress–Displacement Relationship

Figure 10 shows the relationship between the interfacial shear stress–displacement curves (*τ*–*s* curves) and grouting volume under the same roughness. The whole shearing process exhibits no softening behavior, showing a relatively stable trend in the final stage of shearing. Under the same roughness and loading conditions, the peak shear stress gradually increases as the grouting volume increases from 0 to 0.4 dm^3^. The initial stage of almost all the *τ*–*s* curves (except for the curve of specimen R3-V3-C100-S100) becomes steeper with increasing grouting volume, indicating that the initial shear stiffness can be increased by injecting grouts into the interface. Figure 10b shows a slight abnormality in that the curve of specimen R3-V3-C100-S100 is higher than that of specimen R3-V4-C100-S100 during the early stage and middle stage, which may originate from the injection of more grout at the interface rather than in the soil for specimen R3-V3-C100-S100.

#### 3.2.2. Interfacial Shear Strength

Figure 11 shows the fitted curves between peak shear stress and normal stress for cases with different grouting volumes and roughness values. The variations in the apparent cohesion and friction angle with respect to the grouting volume are shown in Figure 12. Under the three roughness values, the apparent cohesion increases nonlinearly with increasing grouting volume. When R = 0 mm, the apparent cohesion increases by 36.4%, 50.9%, and 185.0% when the grouting volume increases from 0 L to 0.2, 0.3, and 0.4 dm^3^, respectively. The same variation rule can be found under R3 and R6 conditions in Figure 12a. The slope of the fitted curves exhibits no obvious changes in Figure 11, and the average value of the friction angle under different grouting volumes changes slightly, ranging from 32.7° to 34.7° at the same roughness in Figure 12b. These characteristics indicate that the interfacial shear strength can be increased by grouting on the interface. The main reason is that the interfacial apparent cohesion increases with the increase of grouting volume, whereas the friction angle shows no significant change.

#### 3.2.3. Shear Dilatancy

Figure 13 displays the variation in cohesive soil dilation or shrinkage with respect to the grouting volume for specimens with R0 in different loading cases. The specific values of the dilation/shrinkage are shown in Table 7. In general, the vertical displacement decreases with increasing grouting volume. Under the C100-S100 loading condition, the specimen exhibits obvious dilatancy, which becomes increasingly weaker with increasing grouting volume, as shown in Figure 13a. Although there exists a descending trend during the late stage under the C100-S75 and C100-S50 loading conditions, the grouted cohesive soil shows significant changes from shrinkage to dilation as the grouting volume varies from 0 to 0.4 dm^3^, as shown in Figure 13b,c. This indicates that the shrinkage of grouted cohesive soil becomes weaker and the dilation becomes more obvious as the grouting volume increases.

### 3.3. Thickness and Distribution of Shear Band

The reasons for the increase in interfacial shear strength can be found from the mesoscale or microscale changes in the shear band; many studies have investigated this phenomenon through numerical analysis. However, through the measurement method proposed in Section 2.3, the results in the current test were obtained in a more accurate and intuitive manner.

Figure 14 shows the variations in the interfacial shear band and apparent soil characteristics along the shearing direction. Along the shearing direction (‘left’ in Figure 14), the soil becomes denser. The interaction between the vertical adjacent soil layers is obvious, and the shear band is thicker. In the direction opposing the shearing direction (‘right’ in Figure 14), the soil becomes loose. This interaction between adjacent soil layers is weak, and even the colored bar is cut horizontally, which results in a thin shear band. These features were reflected in the distribution contour of shear band thickness projected to the horizontal plane for specimens with R0 under no grouting conditions in Figure 15. It shows that the shear band tends to be thicker in the middle left and thinner at both ends, and the thickest position deviates from the center line. Along the Y direction perpendicular to the shear direction on the horizontal plane, the shear band displays a symmetrical distribution with thicker center and thinner ends due to the symmetrical boundary conditions. This also indicates that the representative value of the shear band thickness should avoid selecting the area that is greatly affected by the boundaries, so as to ensure that the selected values can accurately reflect the deformation characteristics of the interface.

It should be noted that grouting into the interface under different conditions may result in different grouting modes [38]. The colored bar, especially in Row 3 facing the grouting hole in Figure 5, may be damaged by the grouts, so the thickness and distribution diagram of the shear band shown in Figure 16 and Figure 17 refers to Row 2 which was not damaged in the grouting process.

Figure 16 shows the relationship of the distribution and thickness of the shear band with respect to roughness in different loading cases. Figure 17 shows the relationship of the distribution and thickness of the shear band with respect to the grouting volume. The results show that the shear band exhibits a parabolic distribution rather than a uniform distribution along the shearing direction, which is consistent with the phenomenon in Figure 14. The peak thickness of the shear band deviates from the central line of the shear box, occurring on the left part. The interfacial shear band becomes thicker as the roughness or grouting volume increases. To enable a comparison of the magnitude of change, the six data points from each shear band are averaged, as shown in Table 8 and Table 9. It can be clearly found that the increase of the roughness and grouting volume can thicken the shear band by a maximum of 13.3%, which contributes to increasing the interfacial shear strength. However, after reaching certain values of grouting volume, additional increases in grouting volume will have marginal diminishing effects on interfacial shear strength.

## 4. Conclusions

To simulate the interfacial shearing between soil and concrete piles with post grouting in tip or sides, the multifunctional interfacial shearing instrument at the Key Laboratory of Geotechnical and Underground Engineering of the Ministry of Education, Tongji University (Shanghai, China) was modified and combined with the self-developed grouting device to carry out 48 sets of direct shear tests on grouted cohesive soil–concrete interfaces with different roughness considering the grouting process. A simple method of punching bars of colored particles into soil was designed to observe and measure interfacial shear band thickness and distribution. The effects of grouting and roughness on the shear characteristics of the cohesive soil–concrete interface were discussed and analyzed from four aspects: the shear stress–displacement relationship, interfacial shear strength, dilatancy, and shear band distribution and thickness. The following conclusions can be drawn:
(1)The shear stress–displacement relationship of the cohesive soil–concrete interface can be effectively improved by increasing the interfacial grouting volume and roughness. The initial shear stiffness and peak shear stress increased with increasing grouting volume and roughness.(2)The improvement in the shear performance of the cohesive soil–concrete interface was mainly due to the interfacial apparent cohesion—rather than the friction angle—being improved by increasing the grouting volume and roughness. Increasing the interfacial grouting volume and roughness had no obvious increasing effects on the friction angle.(3)Cohesive soil without grouting exhibited shear shrinkage that weakened with increasing roughness. As the interfacial grouting volume increased, the shrinkage of the grouted cohesive soil transitioned to dilation. Hence, as the grouting volume increased, the shrinkage became weaker and the dilation became more obvious.(4)Interfacial shear band becomes thicker with increasing grouting volume or roughness. In the direction opposing the shearing direction, the soil became looser, whereas the soil became denser in the shearing direction. The shear band exhibited a parabolic distribution rather than a uniform distribution along the shearing direction, and the shear band thickness was larger in the shearing direction. Increasing the roughness and grouting volume effectively thickened the shear band, which contributed to increasing the interfacial shear strength. However, after reaching certain values of grouting volume, additional increase in grouting volume may have marginal diminishing effects on the interfacial shear strength.


## Figures and Tables

**Figure 1 materials-13-03043-f001:**
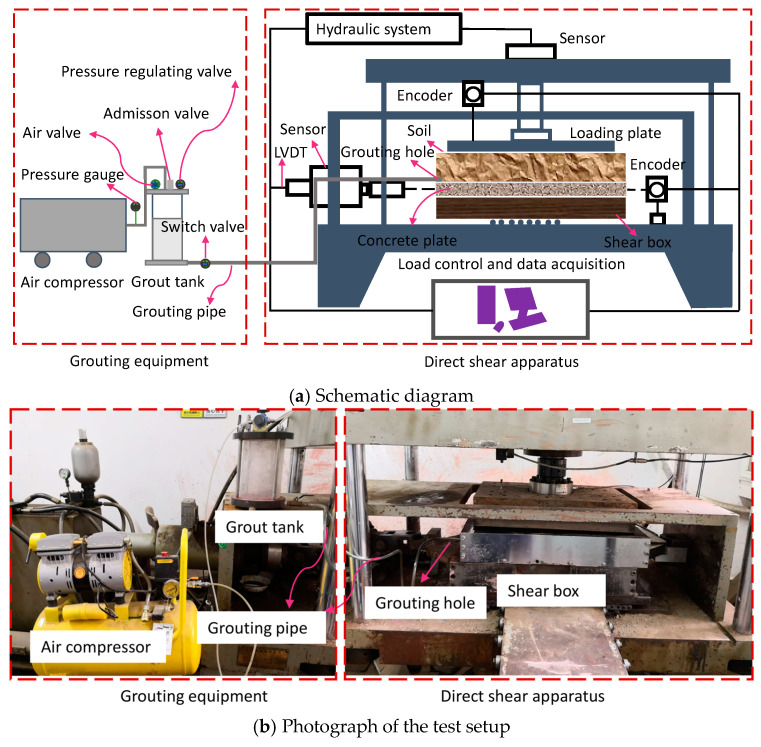
Apparatus for grouting and direct shear test: (**a**) schematic diagram and (**b**) photograph of the test setup.

**Figure 2 materials-13-03043-f002:**
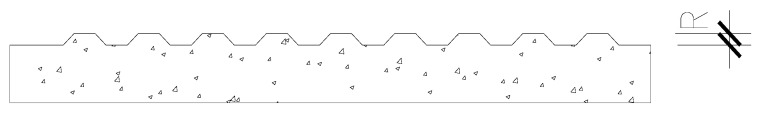
Roughness definition.

**Figure 3 materials-13-03043-f003:**
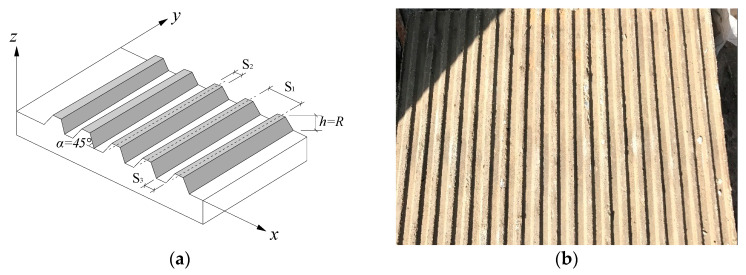
Schematic diagram and photograph of the concrete plate: (**a**) schematic diagram; (**b**) photograph of the concrete plate.

**Figure 4 materials-13-03043-f004:**
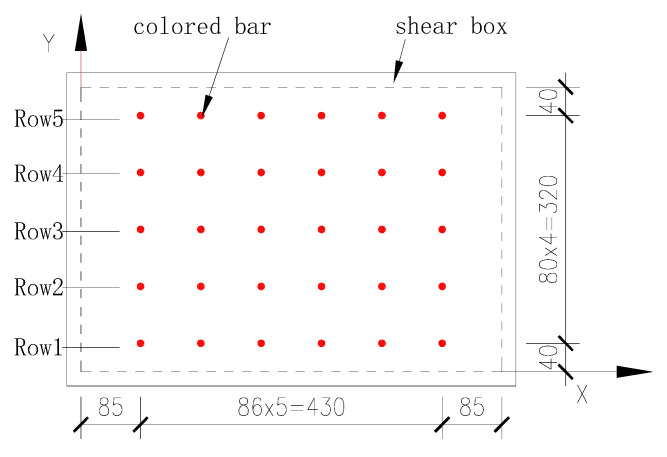
Distribution of colored bars in the upper shear box soil (units: mm).

**Figure 5 materials-13-03043-f005:**
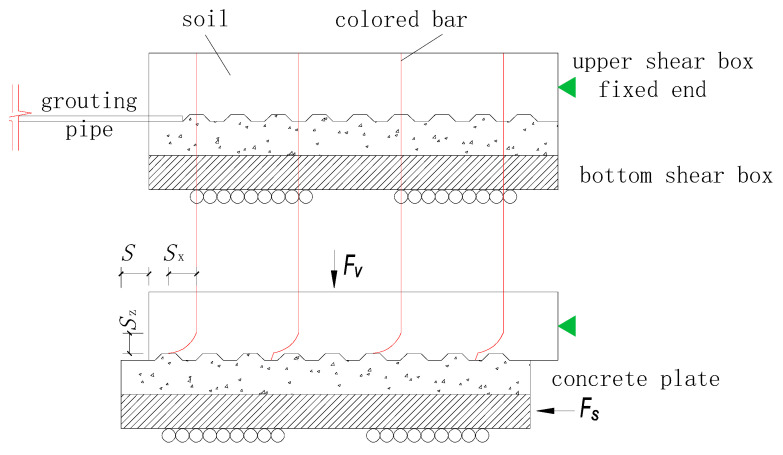
Schematic diagram of the shearing process and formation of the shear band.

**Figure 6 materials-13-03043-f006:**
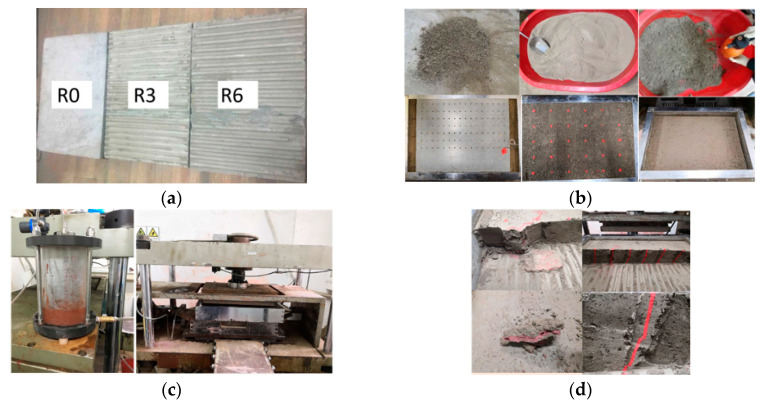
Main test procedures. (**a**) Concrete plate, (**b**) Soil, punching, colored bar, (**c**) Loading, consolidation, grouting, curing, and shearing, (**d**) Cutting, observation, and measurement.

**Figure 7 materials-13-03043-f007:**
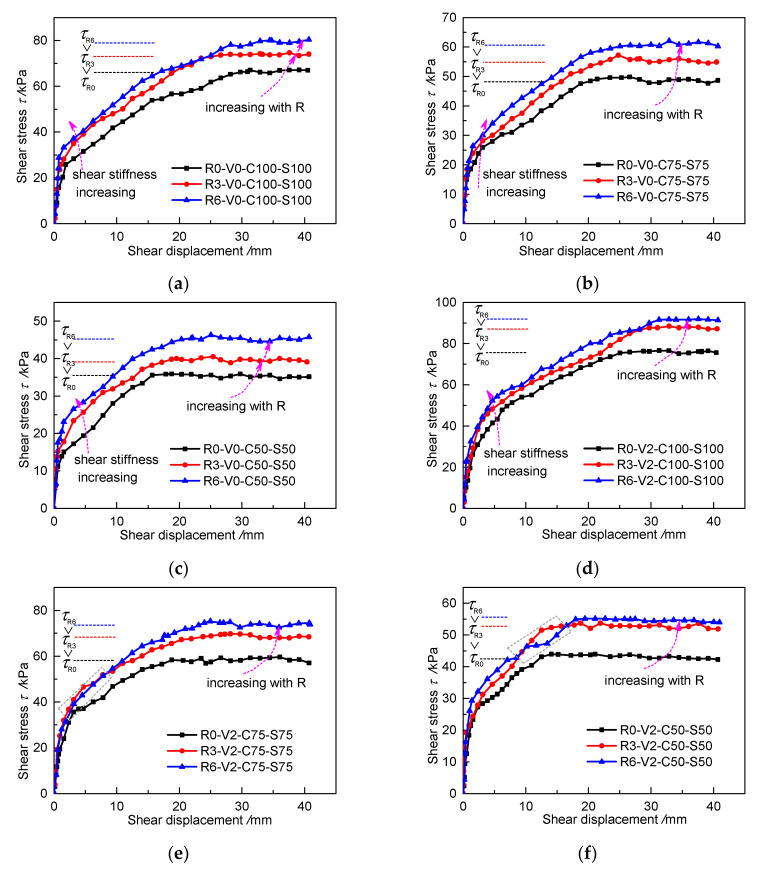
Shear stress–displacement curves (*τ*–*s* curves) for specimens with different roughness values in different cases: (**a**) V0, C100 and S100; (**b**) V0, C75 and S75; (**c**) V0, C50 and S50; (**d**) V2, C100 and S100; (**e**) V2, C75 and S75; (**f**) V2, C50 and S50.

**Figure 8 materials-13-03043-f008:**
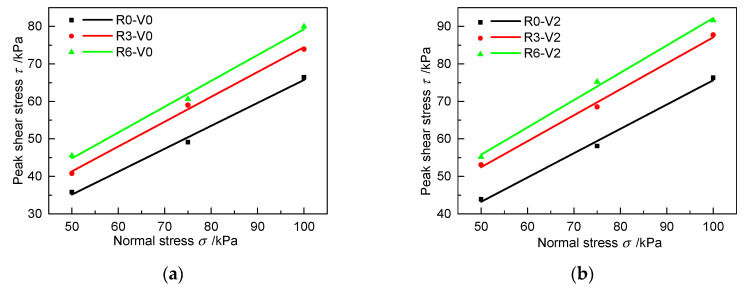
Fitted curves between peak shear stress and normal stress with different roughness values in different grouting cases: (**a**) V0 and (**b**) V2.

**Figure 9 materials-13-03043-f009:**
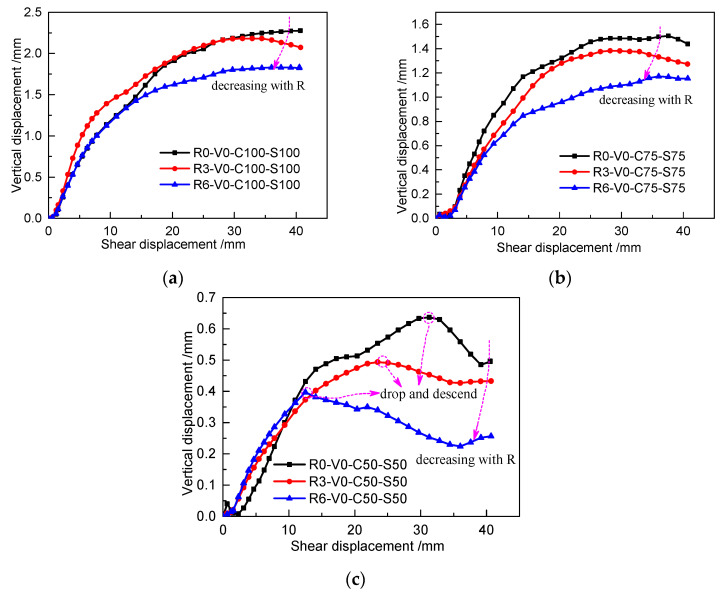
Dilation/shrinkage varying with different roughness values in different loading cases: (**a**) C100 and S100, (**b**) C75 and S75, and (**c**) C50 and S50.

**Figure 10 materials-13-03043-f010:**
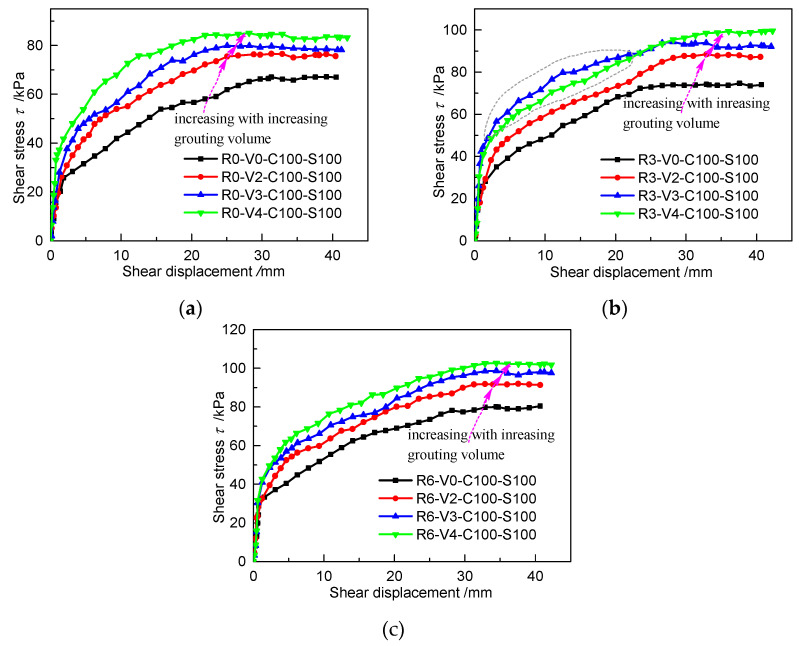
Shear stress–displacement curves (*τ*–*s* curves) for specimens with different grouting volumes: (**a**) R0, C100 and S100; (**b**) R3, C100 and S100; (**c**) R6, C100 and S100.

**Figure 11 materials-13-03043-f011:**
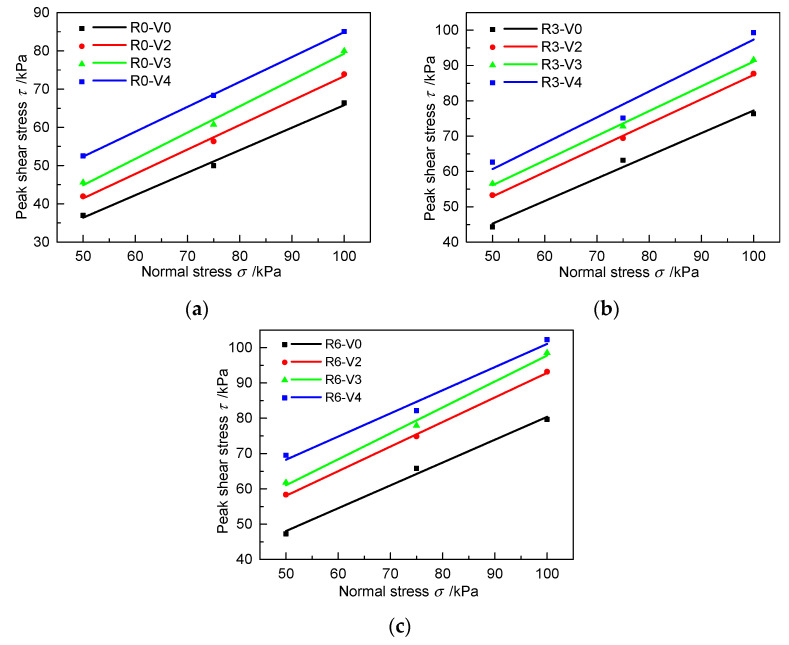
Fitted curves between peak shear stress and normal stress with different grouting volumes for different roughness values: (**a**) R0, (**b**) R3, and (**c**) R6.

**Figure 12 materials-13-03043-f012:**
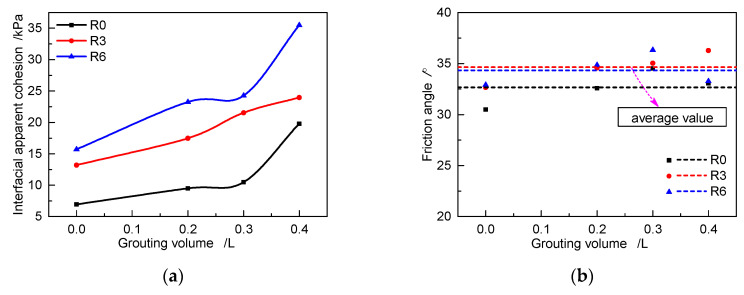
Relationship of interfacial apparent cohesion and friction angle with respect to the grouting volume for specimens with different roughness values under unloading conditions: (**a**) Interfacial apparent cohesion variation with grouting volume; (**b**) Friction angle variation with grouting volume.

**Figure 13 materials-13-03043-f013:**
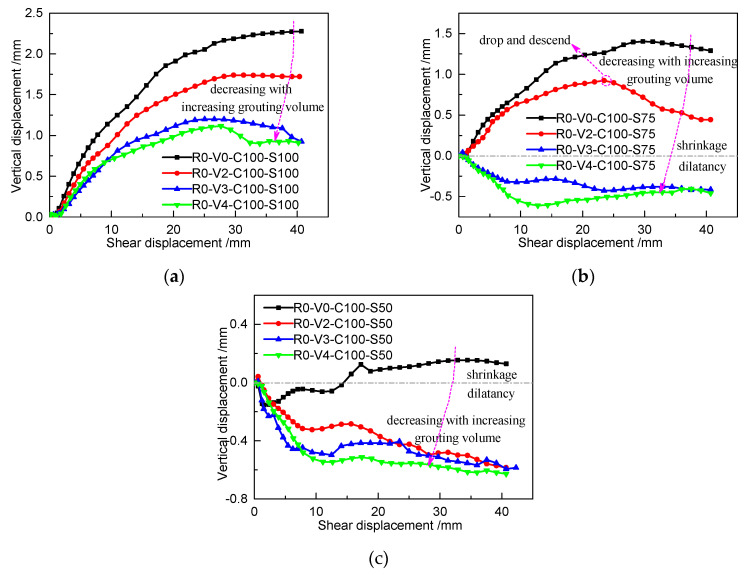
Dilation/shrinkage varying with increasing grouting volume for specimens with R0 in different loading cases: (**a**) C100-S100, (**b**) C100-S75, and (**c**) C100-S50.

**Figure 14 materials-13-03043-f014:**
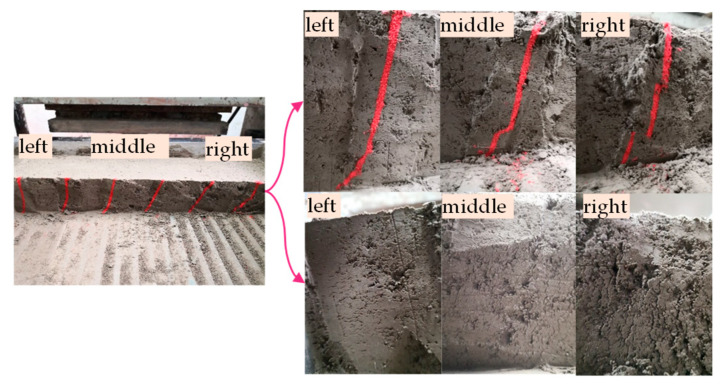
Variations in the shear band and apparent soil characteristics along the shearing direction.

**Figure 15 materials-13-03043-f015:**
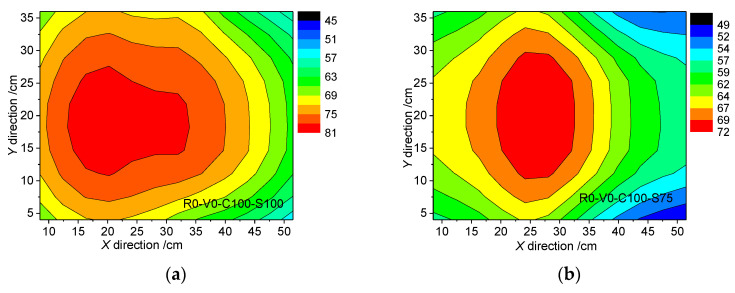
Distribution contour of shear band thickness projected to the horizontal plane: (**a**) R0-V0-C100-S100; (**b**) R0-V0-C100-S75.

**Figure 16 materials-13-03043-f016:**
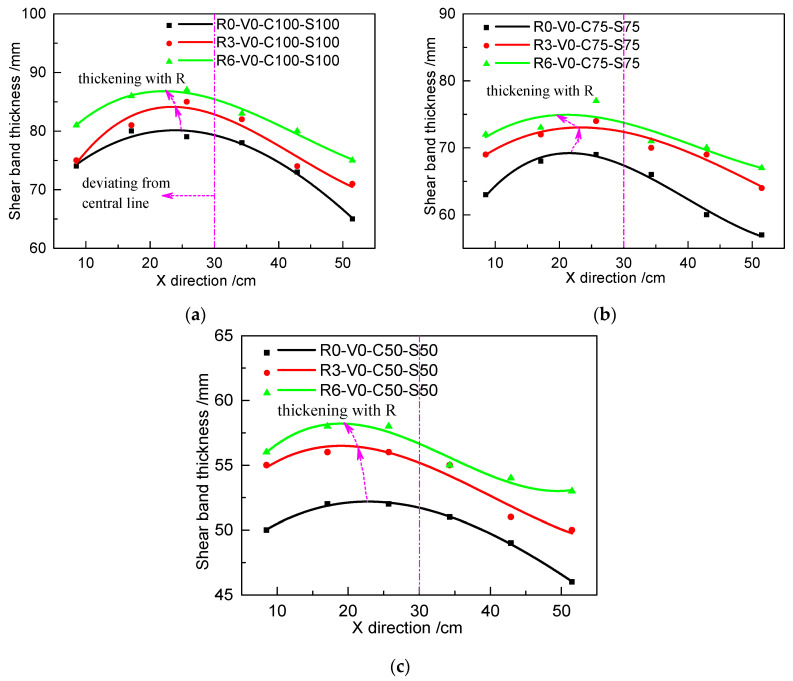
Relationship of the distribution and thickness of the shear band with different roughness values in different loading cases: (**a**) C100 and S100; (**b**) C75 and S75; and (**c**) C50 and S50.

**Figure 17 materials-13-03043-f017:**
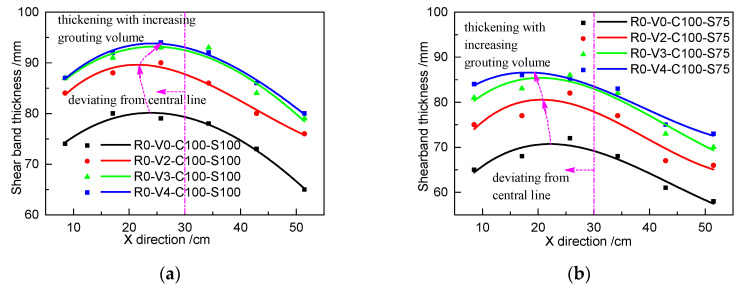
Relationship of the distribution and thickness of the shear band with different grouting volumes in different loading cases: (**a**) C100 and S100 and (**b**) C100 and S75.

**Table 1 materials-13-03043-t001:** Parameters of the SJW-200 direct shear apparatus

Shear Box Size (mm)	Range (kN)		Velocity (mm/min)		Maximum Displacement (mm)	Accuracy (%)
	Normal	Horizontal	Normal	Horizontal		
600 × 400 × 200	200	200	0.1–50	0.1–50	150	0.5

**Table 2 materials-13-03043-t002:** Soil sample parameters

Water Content (%)	Liquid Limit (%)	Plastic Limit (%)	Cohesion (kPa)	Friction Angle (°)	Dry Density (kg/m^3^)
25.0	35.8	21.0	18.52	33.10	1630

**Table 3 materials-13-03043-t003:** Cement parameters.

Constituent	SiO_2_	Al_2_O_3_	Fe_2_O_3_	CaO	MgO	SO_3_	Specific Surface Area (m^2^/kg)
Content	21.50	5.88	3.67	60.2	1.82	2.46	378

**Table 4 materials-13-03043-t004:** Details of test specimens with roughness R0

Specimen	Grouting Volume (dm^3^)	Normal Consolidation Stress (kPa)	Normal Stress When Shearing (kPa)
R0-V0-C100-S100	0	100	100
R0-V2-C100-S100	0.2	100	100
R0-V3-C100-S100	0.3	100	100
R0-V4-C100-S100	0.4	100	100
R0-V0-C100-S75	0	100	75
R0-V2-C100-S75	0.2	100	75
R0-V3-C100-S75	0.3	100	75
R0-V4-C100-S75	0.4	100	75
R0-V0-C100-S50	0	100	50
R0-V2-C100-S50	0.2	100	50
R0-V3-C100-S50	0.3	100	50
R0-V4-C100-S50	0.4	100	50
R0-V0-C75-S75	0	75	75
R0-V2-C75-S75	0.2	75	75
R0-V0-C50-S50	0	50	50
R0-V2-C50-S50	0.2	50	50

Note: In specimen ‘R0-V0-C100-S100’, ‘R’ represents roughness (‘R0’, ‘R3’, and ‘R6’ represent roughness values of 0, 3, and 6 mm, respectively), ‘V’ represents grouting volume (‘V0’, ‘V2’, ‘V3’, and ‘V4’ represent grouting volumes of 0, 0.2, 0.3, and 0.4 dm^3^, respectively), ‘C’ represents normal consolidation stress, and ‘S’ represents normal stress during shearing.

**Table 5 materials-13-03043-t005:** Fitted curve parameters (interfacial apparent cohesion and friction angle) conforming to Mohr–Coulomb shear criterion for specimens with different roughness values

Grouting Volume	Roughness	Apparent Cohesion *c*_e_ (kPa)	Friction Angle *φ*_e_ (°)	Correlation Coefficient R^2^
V0	R0	4.61	31.42	0.989
	R3	8.26	33.50	0.993
	R6	10.4	34.52	0.990
V2	R0	10.83	32.94	0.989
	R3	17.86	34.68	0.992
	R6	19.39	36.05	0.993

**Table 6 materials-13-03043-t006:** Final dilation/shrinkage for specimens with different roughness values in loading conditions without grouting.

Grouting Volume	Loading Conditions	Roughness	Dilation/Shrinkage (mm)	Variation (%)
V0	C100-S100	R0	2.277	-
		R3	2.073	−9.0
		R6	1.823	−12.1
	C75-S75	R0	1.438	-
		R3	1.272	−11.5
		R6	1.154	−9.3
	C50-S50	R0	0.497	-
		R3	0.433	−12.9
		R6	0.257	−40.6

Note: A positive value corresponds to ‘shrinkage’ in the fourth column, whereas a negative value corresponds to ‘dilation’.

**Table 7 materials-13-03043-t007:** Final dilation/shrinkage for specimens with R0 and different grouting volumes under unloading conditions

Roughness	Loading Conditions	Grouting Volume	Dilation/Shrinkage (mm)	Relative Variation (%)
R0	C100-S100	V0	2.277	-
		V2	1.722	−24.4
		V3	0.924	−46.3
		V4	0.910	−1.5
	C100-S75	V0	1.288	-
		V2	0.443	−65.6
		V3	−0.414	−193.6
		V4	−0.458	−10.6
	C100-S50	V0	0.128	-
		V2	−0.584	−556.3
		V3	−0.594	−1.7
		V4	−0.627	−5.6

Note: ‘Shrinkage’ corresponds to positive values, whereas ‘dilation’ corresponds to negative values in the fourth column.

**Table 8 materials-13-03043-t008:** Average shear band thickness for specimens with different roughness values and no grouting

Loading Conditions	Roughness	Average Shear Band Thickness (mm)	Relative Variation (%)
C100-S100	R0	74.8	-
	R3	78.0	4.3
	R6	82.0	5.1
C75-S75	R0	63.8	-
	R3	69.7	9.2
	R6	71.7	2.9
C50-S50	R0	50.0	-
	R3	53.8	7.6
	R6	55.6	3.3

**Table 9 materials-13-03043-t009:** Average shear band thickness for specimens with R0 under different grouting conditions

Loading Conditions	Grouting Volume	Average Shear Band Thickness (mm)	Relative Variation (%)
C100-S100	V0	74.8	-
	V2	84.0	12.3
	V3	87.7	4.4
	V4	88.5	0.9
C100-S75	V0	65.3	-
	V2	74.0	13.3
	V3	79.2	7.0
	V4	81.0	2.3

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
