# Peer review of "Study on the Effects of Grouting and Roughness on the Shear Behavior of Cohesive Soil–Concrete Interfaces"

_materials, 2020, doi:10.3390/ma13143043_

Round 1

Reviewer 1 Report

This manuscript describes an experimental study on the 48 groups of direct shear tests to study the effects of grouting and roughness on the shear behavior of cohesive soil-concrete interfaces. The topic of the work is very interesting and relevant to the material engineering community.  The authors' study focuses on the grouting material and roughness interface. While the topic is interesting, this reviewer believes that the manuscript is not suitable for publication at this stage. The authors present a series of discussions surrounding the different measurement locations/types that need to be revisited. The authors should enlarge the bibliographic context by referring to papers from the literature addressing additional problems and methodologies about the testing of the interface.

In most cases, the authors seem to draw conclusions from each of the measurement locations, but again this is a limited study. 

The introduction does not identify the knowledge gap that the authors are trying to address with their research. The introduction lists some relevant research but fails to present a scientific review. Please consider rewriting and clarify the motivation, objectives, and significance of this study. The objective statements are rather vague and lack projected outcomes or how the paper will assist practitioners.

The aspects of the novelty of the paper must be pointed out more clearly, particularly in the abstract and concluding remarks sections because the analysis of the data presented in the paper is already presented by past research.

The manuscript needs a thorough review of language and technical writing issues. To this end, the author is encouraged to prepare a revised version of the paper in which the above issues should be considered.

Reviewer 2 Report

The manuscript describes the effect of grouting on shear behavior between the rough concrete surface and the soil. Its originality lies in the combined effect of grouting and roughness of the concrete surface. In this regard, it is not only cognitive, but also practical in underground construction, where precast concrete elements are used.

In general, I rate the article highly, I have no significant comments on its content. Maybe the research could have been carried out for a broader spectrum of soil. This issue can be completed in future studies.

Among the less important comments I could mention:

- Lack of some spaces between text and brackets with reference number (e.g. in lines 52, 55, 56 etc.).

- Grout volume given in liters (e.g. Tab. 4, text line 181 etc). According to MDPI Materials instruction for authors SI units should be used.

- Line 137 – what is the meaning of ⑧?

- The quality of figures 1 and 6 could be improved.

Round 2

Reviewer 1 Report

The authors have made a considerable effort to revise the manuscript, and the main comments about the scientific approach have been successfully addressed. The paper has been significantly improved; however, there are several grammatical mistakes throughout the paper, and English requires substantial editing to correct these mistakes and improve the quality of the written presentation using appropriate scientific English language. Also, there are many parts that need to be written concisely, and revisions and refinements are required throughout the paper. This is important to ensure that the methodology and the findings of this work are successfully conveyed.
